# Dynamics of Tropical Forest Twenty-Five Years after Experimental Logging in Central Amazon Mature Forest

**Márcio R. M. Amaral** [1,*] **, Adriano J. N. Lima** [1] **, Francisco G. Higuchi** [2] **, Joaquim dos Santos** [1] **and Niro Higuchi** [1]

1   Forest Management Laboratory (LMF), National Institute of Amazonian Research (INPA), Manaus 69060-001, Amazonas, Brazil; adriano@inpa.gov.br (A.J.N.L.); joca@inpa.gov.br (J.d.S.); niro@inpa.gov.br (N.H.)
2   HDOM Engenharia e Projetos Ambientais, Forest Management Laboratory (LMF), National Institute of Amazonian Research (INPA) Manaus 69060-001, Amazonas, Brazil; fghiguchi@hdom.com.br
*   Correspondence:Correspondence: marcius.rogerius@gmail.com; Tel.: +55-92-98432-8807

**Abstract:** Long-term studies of the dynamics of managed forests in tropical regions are lacking. This study aimed to evaluate the dynamics of a tropical forest, over a 25-year period, that was experimentally logged in 1987 and 1988 and submitted to three different cutting intensities. All trees with diameter at breast height (DBH) $\geq$ 10 cm have been measured annually since 1990. The three logging intensities that were applied were: light (T1)-trees harvested with DBH $\geq$ 55 cm; medium (T2)-DBH $\geq$ 50 cm; and heavy (T3)-DBH $\geq$ 40 cm. Control plots (T0) were also monitored. The highest mean annual mortality rates (1.82% $\pm$ 0.38), recruitment rates (2.93% $\pm$ 0.77) and diameter increments (0.30 $\pm$ 0.02 cm) occurred in the T3 treatment. Shifts in dynamics of the forest were mainly caused by a striking increase in a fast-growing pioneer species and their high mortality rates. The loss in stocking caused by mortality was greater than to that of replacement by recruitment. The results demonstrated that selective logging altered the natural dynamics of the forest through increased: mortality rates, recruitment and growth rates of the residual trees.

**Keywords:** forest dynamics; tree mortality; tree recruitment; tree Growth; tropical forests; logging; silvicultural treatments

## 1. Introduction

The Amazon forest is known as the largest tropical forest in the world [1,2], and is characterized by its different types of vegetation and elevated floristic diversity [3–5]. Aside from the conservation of its biodiversity, the maintenance of this biome is paramount for nutrient cycling, the hydrological cycle and the global carbon pool. The Amazon forest has an estimated carbon pool between 70 and 80 billion metric tons [6,7], and through photosynthesis and respiration, approximately 18 billion metric tons are sequestered annually [8].

Although there is a consensus on the importance of the Amazon forest for the maintenance of ecosystem services, the constant pressures and the abuse of forest resources threaten its conservation. The main pressures from land uses are linked to intensive livestock farming, agriculture and poorly planned logging operations. The process of forest degradation often begins with poorly planned selective logging, of just a few tree species, also known as "high grading." These logging operations target high value trees that are more attractive to the timber market. After depleting the stocking of the previously forested site, the remaining vegetation is replaced by non-productive pasture and/or agricultural crops [9], followed by land abandonment. This dynamic of deforestation has led to

an increase in the concern and urgency for the protection of the remaining natural areas in the Amazon, through the adoption of systematic conservation planning.

One of the alternatives for the maintenance of Amazonian forests is through Sustainable Forest Management (SFM). Unlike other forms of land use practiced in the Amazon and other tropical regions, SFM is currently thought to be one of the best alternatives for the sustainable use of forest resources. In addition to generating social and economic benefits, SFM contributes to the maintenance and guarantee of forest stocking, conservation of biodiversity, removal of carbon dioxide from the atmosphere and contributes to land use planning [10–13].

However, during logging activities, the residual trees are still damaged [14], which is due to the tree felling and removal of commercial timber. On average, for each logged tree there is the loss of four residual trees and severe damage to approximately 0.5 m$^2$ of basal area in stems 10 cm and greater at diameter at breast height (DBH) [15]. An experiment of forest management studying different logging intensities in Central Amazonia (BIONTE Project) demonstrated that the ratio between the logged volume and volume of damaged or dead trees is 1:1. This ratio of dead and damaged trees was then subdivided into the following ratios: 1:13, 1:11 and 1:7 for light, moderate and heavy logging intensity, respectively [16].

The damage to the residual forest influences the quality of the forest, compromising its productivity and natural regeneration. Some studies on the impacts of logging in Amazon rainforests have been extensively investigated, although these studies have been limited to post-logging intervals of only several decades [11,17,18]. However, literature on long-term effects to structural dynamics, species composition and the yield of tropical forests are lacking [10,11,17,19,20]. This is especially the case when it comes to recovering the stocking of commercial species, which are the main target of timber extraction.

The data from managed forests are paramount to understanding the effects of logging on forest structure, function and species composition [14]. In the same way, the understanding of forest dynamics is fundamental for conducting, planning and establishing strategies for forest management. Monitoring the dynamics of the remaining forest, especially in the long term, can help to understand if tropical forests are actually resilient to selective logging [17]. Additionally, the results may indicate whether the quantity and quality, in terms of the maintenance of the commercial species, of the forest is maintained. Moreover, this information could contribute to an understanding of various ecological processes that include patterns of change, maintenance of species richness [21,22] and the complexity of post-harvest plant communities [23].

Systematic and long-term monitoring are essential to accompany the dynamics and ecological processes of forests [24], such as mortality, recruitment, turnover and growth [25]. In this sense, the objective of this work was to study the dynamics (recruitment, mortality and increment) of a selectively logged mature tropical forest, with different logging intensities, in Central Amazonia. Our study sought to answer if there is a difference between the different logging intensities on the forest dynamics with the passage of time. In addition, we sought to determine which species had the highest recruitment, the greatest mortality and the highest growth rate in response to each logging intensity. This work was conducted at the Experimental Station of Tropical Forestry (E.E.S.T./ZF2) of the National Institute of Amazonian Research (INPA).

## 2. Materials and Methods

### 2.1. Study Area

The study was conducted at the Experimental Station of Tropical Forestry (EEST/ZF2) of the National Institute of Amazonian Research (INPA), located north of Manaus in the state of Amazonas, Brazil (approximate coordinates: 02°37′ S to 02°38′ S and 60°09′ W to 60°11′ W). The vegetation of the studied site is characterized as a humid tropical forest, heterogenic with high floristic diversity [26–28]. The soils of the region are poor in nutrients and the topography is characterized by smooth plateaus with three different toposequences consisting of valley bottoms, slopes and plateaus [12]. According

to Köppen's classification, the climate of the EEST is a tropical climate of the type Af [29], characterized by high temperatures and humidity. The amplitude of yearlong temperature, precipitation and relative humidity is: 34.7 °C and 20.8 °C, 116.2 mm and 00.0 mm and 100% and 69.0%, respectively [30].

## 2.2. Experimental Desingn

The original experiment began in 1979 and was titled "Ecological Management and Exploration of the Tropical Rainforest". The initial objective of this project was to evaluate the potential of wood production and present a model of forest management for the Central Amazon. The project was subdivided into two subprojects: (i) Ecology and Forest Management; and (ii) Forest Products Technology.

In 1980, four blocks (600 × 400 m) of 24 hectares each were established. These blocks were then subdivided into 4 hectare (200 × 200 m) sub-blocks. In the center of each sub-block a 1 hectare (100 × 100 m) permanent plot (PP) was established. These PP's were installed in the center of the sub-blocks to minimize "Edge Effect" of other the treatments [31]. In each sub-block, three treatments of differing logging intensities were applied (T1, T2 and T3) as well as a control (T0). This was then replicated three times. Logging operations began in 1987 (Table 1).

**Table 1.** Silvicultural treatment intensity, trees removed and volume removed. Details: TTS: treatments; INT: Intensity; LY: logging year; ML: Minimum diameter at breast height (DBH) Logging; RMV: average of trees removed; VOL: average volume removed; BA: average basal area removed.

| TTS | INT | LY | ML | RMV $(Tree^{-1} \cdot ha^{-1})$ | VOL $(m^3 \cdot ha^{-1})$ | BA $(m^2 \cdot ha^{-1})$ |
|-----|-----|-----|-----|-----|-----|-----|
| $T_0$ | Control | - | - | - | - | - |
| $T_1$ | Low Intensity | 1987 | $\geq 55$ | 5 | 34 | 32% |
| $T_2$ | Medium Intensity | 1987 | $\geq 50$ | 8 | 49 | 42% |
| $T_3$ | Heavy Intensity | 1988 | $\geq 40$ | 16 | 67 | 69% |

The initial forest inventory was conducted in 1980. The assessment's objectives were to quantitatively and qualitatively estimate commercial volume stocking. This initial forest inventory included all trees with a diameter at breast height (DBH) of 25 cm and greater with the identification of the species. From 1986 onwards to the most current inventory, the minimum measurable DBH was established at 10 cm and greater. The remeasuring of the permanent plots has been conducted annually since 1990, every July. Logging operations concentrated on commercial species, which were determined based on the current market at the time of operations. The forest inventory of this study identified 50 target species [28].

## 2.3. Data Collection

Data for this study were derived from 25 years (1990 to 2015) of forest inventory from the 12 permanent plots of 1 hectare each. The DBH for all trees ($\geq 10$ cm) were measured. Each tree was identified by its common name by experienced parataxonomists. Afterwards, dendrological collections of each individual tree were taken and conducted at INPA's herbarium for botanical identification to the species level. In every monitored year, additional trees were added to the inventory (recruitment), after attaining the minimum DBH of 10 cm, while the trees that died (mortality) were excluded from the inventory.

*2.4. Data Analysis*

2.4.1. Biomass

Estimate of total (above- and belowground) individual Fresh Biomass was based on the nonlinear model proposed by Silva, 2007 [32] ($R^2$ = 0.94 and Syx% = 2.02):

$$\text{Biomass} = 2.7179 \times \text{DBH}^{1.8774} \tag{1}$$

where Biomass is the fresh biomass in kg; DBH is the diameter at breast height in cm;

2.4.2. Bole Volume

Estimate of bole volume was calculated based on site-specific volumetric equations develop by [33] ($R^2$ = 0.89 and Syx% = 2.02):

$$\text{Volume} = 0.001176 \times \text{DBH}^{1.99868} \tag{2}$$

where Volume is the volume in m$^3$; DBH is the diameter at breast height in cm;

2.4.3. Mortality Rate

Mortality rate was determined by counting absolute number of dead trees within the monitoring period and corresponding number of trees measured:

$$\text{MT} = \left(\frac{M_{t+1}}{N_t}\right) \times 10 \tag{3}$$

where MT = mortality rate; $M_{(t+1)}$ = number of trees registered as "dead" in the monitoring period "t + 1"; $N_t$ = total number of measured trees in monitoring period "t";

2.4.4. Recruitment Rate

Recruitment rate was calculated by counting absolute number of recruit trees within the monitoring period and corresponding number of trees measured:

$$\text{RT} = \left(\frac{R_{t+1}}{N_t}\right) \times 100 \tag{4}$$

where RT = recruitment rate; $R_{(t+1)}$ = number of trees registered as "recruits" in the monitoring period "t + 1"; $N_t$ = total number of measured trees in monitoring period "t";

2.4.5. Individual Increment

For individual diameter increment monitoring, only tree species that presented at least three individuals, in each treatment and three monitored DBH measures within the monitoring period were selected.

$$\text{IDi} = \text{IDi}_{t+1} - \text{IDi}_t \tag{5}$$

where ID = diameter increment of species i; $ID_{t+1}$ = diameter increment of species *i* in time "t + 1"; $ID_t$ = diameter increment of species i in time "t";

*2.5. Statistical Analysis*

Differences in the mortality rate, recruitment and diameter increment were evaluated with Analysis of Variance (ANOVA) of repeated measurements [34]. In order to overcome the problem of the violation of the principle of independence of data, as these are permanent plots and remeasurement is always of the same individuals, it was necessary to correct the value of "*F*" to each source of variation.

Therefore, Greenhouse-Geisser correction was used. When the ANOVA test presented significant statistical results ($p < 0.05$), a post hoc test of Tukey was applied. Linear regressions were also adjusted for diameter increment across the years for all remaining trees of each treatment. In addition, a *t*-test was performed between the recruitment and mortality stocks of each treatment and presented in Figure 3. All analysis were performed on Systat 13.2 free versions.

## 3. Results

### 3.1. Mortality Rates

During the 25-year monitoring period, the mean number of dead trees per hectare per treatment during this period was $156 \pm 7$ trees·ha$^{-1}$ ($6 \pm 0.3$ tree·ha$^{-1}$·year$^{-1}$) in T0, $269 \pm 42$ trees·ha$^{-1}$ ($11 \pm 2$ tree·ha$^{-1}$·year$^{-1}$) in T1, 230 $\pm$ 56 trees ha$^{-1}$ ($09 \pm 2$ tree·ha$^{-1}$·year$^{-1}$) in T2 and 280 $\pm$ 17 trees ha$^{-1}$ ($12 \pm 1$ tree·ha$^{-1}$·year$^{-1}$) in T3. These values represented a higher annual average mortality rate per hectare in T3, T1, followed by T2. Despite the fact that T1 subjected to low intensity logging, the mortality rate was higher in this treatment than mortality rates in the medium intensity logging. The highest losses in stocking (number of trees, basal area, volume and biomass) were observed in the logged treatments, principally in treatments T1 and T3 (Table 2).

**Table 2.** Total number of trees (*N*) Recruitment Rate and Mortality Rate (RT), Basal Area (BA); Volume (VOL); Biomass (BIO), Carbon (CAR) that came out of the system through mortality and recruitment, respectively, over 25 years of observation.

| Treatments | | *N* | RT (%) | BA (m²·ha⁻¹·year⁻¹) | VOL (m³·ha⁻¹·year⁻¹) | BIO (t·ha⁻¹·year⁻¹) | CAR (t·ha⁻¹·year⁻¹) |
|---|---|---|---|---|---|---|---|
| T0 | | 469 | $0.97 \pm 0.14$ | $0.30 \pm 0.06$ | $4.49 \pm 0.82$ | $6.78 \pm 1.20$ | $1.92 \pm 0.34$ |
| T1 | Mortality | 806 | $1.58 \pm 0.30$ | $0.41 \pm 0.09$ | $6.08 \pm 1.35$ | $9.39 \pm 2.08$ | $2.66 \pm 0.59$ |
| T2 | | 691 | $1.34 \pm 0.21$ | $0.37 \pm 0.13$ | $5.53 \pm 1.91$ | $8.49 \pm 2.88$ | $2.40 \pm 0.82$ |
| T3 | | 840 | $1.82 \pm 0.38$ | $0.46 \pm 0.09$ | $6.89 \pm 1.33$ | $10.59 \pm 2.01$ | $3.0 \pm 0.57$ |
| T0 | | 385 | $0.85 \pm 0.18$ | $0.05 \pm 0.01$ | $0.73 \pm 0.15$ | $1.26 \pm 0.26$ | $0.36 \pm 0.07$ |
| T1 | Recruitment | 1146 | $2.54 \pm 0.67$ | $0.15 \pm 0.04$ | $2.31 \pm 0.61$ | $3.97 \pm 1.05$ | $1.12 \pm 0.30$ |
| T2 | | 1009 | $2.17 \pm 0.58$ | $0.13 \pm 0.04$ | $2.00 \pm 0.59$ | $3.44 \pm 1.02$ | $0.97 \pm 0.29$ |
| T3 | | 1298 | $2.93 \pm 0.77$ | $0.18 \pm 0.05$ | $2.62 \pm 0.73$ | $4.51 \pm 1.26$ | $1.28 \pm 0.36$ |

The ANOVA Repeated Measures for mortality rates over the 25 year monitoring period, showed that the mortality rates differed statistically ($p < 0.002$). However, between the control treatment (T0) and the medium logging intensity treatment (T2), there were no significant differences between mortality rates (Tukey post hoc, $p > 0.09$). Comparing only those treatments where there was a reduction of basal area from logging, the results were not significant between T1 and T2 (Tukey post hoc, $p = 0.338$) and T1 and T3 (Tukey post hoc, $p = 0.451$). However, mortality rates between the moderate (T2) and heavy (T3) logging treatment showed lower T2 values (Tukey post hoc, $p = 0.040$).

The mortality varied significantly over the years of monitoring, in all treatments analyzed (ANOVA of Repeated Measures, $p < 0.0001$). These values were more apparent shortly after logging, reaching mortality rates of 2.4% in T1, 2.4% in T2 and 4.6% in T3, decreasing and stabilizing over the years (Figure 1). However, in the final years of monitoring, an increase in the mortality rate in all treatments reoccurred. This increase in the mortality rate was more significant in the treatments where there was logging, and in some cases, these rates of mortality exceeded the rates found after the initial logging operations (T1 and T2).

The peak in the mortality rate for each treatment was: 3.8% for T1 in 2008; 2.5% for T2 in 2010; and 4.6% for T3 in 1990. For the treatment with heaviest logging (T3), in 2010 the mortality rate was also significant (4.1%). These results show that tree mortality was higher in treatments where there was logging, when compared to the control treatment (T0).

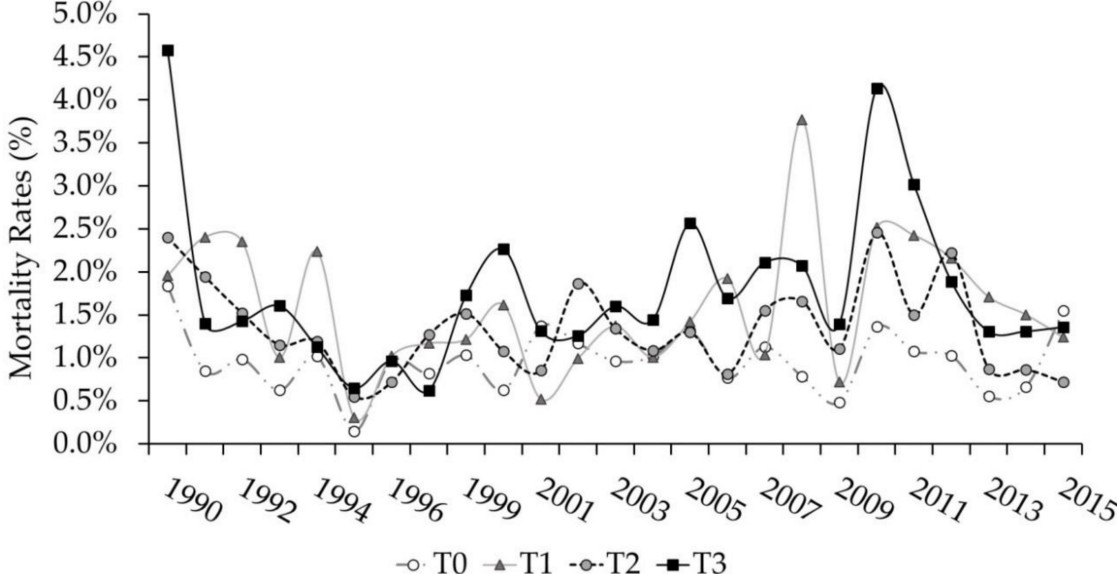

**Figure 1.** Dynamics in the mortality rate observed over 25 years of monitoring for all treatments studied.

In floristic terms, the botanical family that presented the highest number of dead trees, in all treatments where there was a reduction of basal area, was Urticaceae, with 13.2% in T1; 12.9% in T2; and 20.6% in T3. However, the botanical family that demonstrated the highest losses to stocking was Fabaceae, in all of the treatments studied. The treatment with greatest loss in stocking was the heavy intensity, followed by the low intensity and medium intensity. In the control treatment (T0) the botanical family with the greatest mortality was Fabaceae at 12.4%. For this same treatment, the Urticaceae family represented a value of only 4.3% of the total mortality.

The species *Cecropia sciadophylla* Mart., from the Urticaceae family, showed the highest number of dead trees in all logged treatments, with a reduction in basal area of 8.3% in T1, 8.0% in T2 and 14.4% in T3. This species represented an output of 6.2%, 5.7% and 9.7% of the total forest biomass, respectively for T1, T2 and T3 treatments. In the control treatment (T0) the species *Protium* spp. (9.4%), had the greatest loss in forest biomass (4.9%). In Table 3, it is possible to observe the five main species, of each treatment analyzed, that most contributed to the reduction in basal area, volume, biomass and forest carbon.

**Table 3.** Species with the highest total number of: dead trees (*N*); Basal Area loss (BA); Volume (VOL); Biomass (BIO); Carbon (CAR) and Ecological Group (EG): No Pioneer (NP), and Pioneer Species (PS).

| Treatment | Species | Botany Family | *N* | BA (m$^2$) | VOL (m$^3$) | BIO (t) | CAR (t) | EG |
|---|---|---|---|---|---|---|---|---|
| | *Protium* spp. | Burseraceae | 44 | 1.0 | 15.6 | 25.1 | 7.1 | NP |
| | *Eschweilera* spp. | Lecythidaceae | 22 | 0.8 | 11.9 | 18.8 | 5.3 | NP |
| T0 | *Lauraceae* spp. * | Lauraceae | 16 | 0.5 | 6.9 | 11.0 | 3.1 | NP |
| | *Rinorea paniculata* (Mart.) Kuntze | Violaceae | 15 | 0.3 | 3.9 | 6.4 | 1.8 | NP |
| | *Mabea* spp. | Euphorbiaceae | 14 | 0.2 | 3.1 | 5.1 | 1.5 | NP |
| | *Cecropia sciadophylla* Mart. | Urticaceae | 67 | 1.8 | 27.3 | 43.6 | 12.4 | PS |
| | *Vismia* spp. | Hypericaceae | 46 | 0.7 | 10.1 | 17.0 | 4.8 | PS |
| T1 | *Eschweilera* spp. | Lecythidaceae | 42 | 2.0 | 29.5 | 45.3 | 12.8 | NP |
| | *Protium* spp. | Burseraceae | 40 | 0.8 | 12.3 | 20.0 | 5.7 | NP |
| | *Croton urucurana* Baill. | Euphorbiaceae | 39 | 0.7 | 10.4 | 17.1 | 4.9 | PS |
| | *Cecropia sciadophylla* Mart. | Urticaceae | 55 | 1.5 | 22.7 | 36.3 | 10.3 | PS |
| | *Protium* spp. | Burseraceae | 43 | 1.3 | 19.0 | 30.0 | 8.5 | NP |
| T2 | *Croton matourensis* Aubl. | Euphorbiaceae | 30 | 0.8 | 11.5 | 18.4 | 5.2 | PS |
| | *Inga* spp. | Fabaceae | 28 | 0.8 | 12.5 | 19.7 | 5.6 | NP |
| | *Vismia* spp. | Hypericaceae | 26 | 0.4 | 5.5 | 9.3 | 2.6 | PS |
| | *Cecropia sciadophylla* Mart. | Urticaceae | 121 | 3.1 | 45.8 | 73.9 | 20.9 | PS |
| | *Protium* spp. | Burseraceae | 44 | 1.1 | 16.8 | 27.0 | 7.6 | NP |
| T3 | *Miconia* spp. | Melastomataceae | 40 | 0.9 | 13.7 | 22.1 | 6.3 | PS |
| | *Vismia* spp. | Hypericaceae | 36 | 0.6 | 8.8 | 14.6 | 4.1 | PS |
| | *Eschweilera* spp. | Lecythidaceae | 34 | 1.6 | 23.3 | 36.0 | 10.2 | NP |

* Species that was not identified at the genus level but belong to the respective botanical family.

## 3.2. Recruitment Rates

During the monitoring period, the mean number of trees that entered the inventory was $134 \pm 31$ trees·ha$^{-1}$ ($5 \pm 1$ tree·ha$^{-1}$·year$^{-1}$) in T0, $406 \pm 22$ trees·ha$^{-1}$ ($16 \pm 1$ tree·ha$^{-1}$·year$^{-1}$) in T1, $339 \pm 86$ trees· ha$^{-1}$ ($14 \pm 3$ tree·ha$^{-1}$·year$^{-1}$) in T2 and $444 \pm 8$ trees·ha$^{-1}$ ($18 \pm 0.3$ tree·ha$^{-1}$·year$^{-1}$) in T3. The greatest gain in stocking (number of trees, basal area, volume and biomass), through recruitment, occurred in treatments where there was logging, mainly T1 and T2 (Table 2).

When comparing the recruitment rates (ANOVA of repeated measurements), between each treatment analyzed, significant differences were observed in the values obtained ($p < 0.001$). The highest recruitment rates were observed mainly in treatments where there was a reduction of basal area, in which there were probably no significant differences between T1 and T2 (Tukey post hoc, $p = 0.363$) and T1 and T3 (Tukey post hoc, $p = 0.317$). However, recruitment rates between T2 and T3 were statistically different (Tukey post hoc, $p = 0.029$). These results demonstrate that the recruitment of trees was higher in the treatments in which there was reduction of basal area, when compared with the control treatment (T0).

During the analysis period, it was observed that the recruitment rate varied significantly in all treatments (ANOVA of Repeated Measures, $p < 0.0001$). It was also observed that there was a significant effect of the different treatments in the recruitment of trees, that is, there was an interaction Recruitment * Treatments ($p = 0.001$). Soon after logging, the recruitment rates also increased, remaining high for more than 10 years. However, after this period recruitment rates declined considerably beginning in 2003 and stabilized in the following years (Figure 2). The peak recruitment rate for each treatment was: 6.3% for the T1 treatment in 1997; 5.3% for the T2 treatment in 2002; and 7.5% for the T3 treatment in 1993. For the control treatment (T0), the highest peak in the recruitment rate of trees occurred in 1990 at 1.8%.

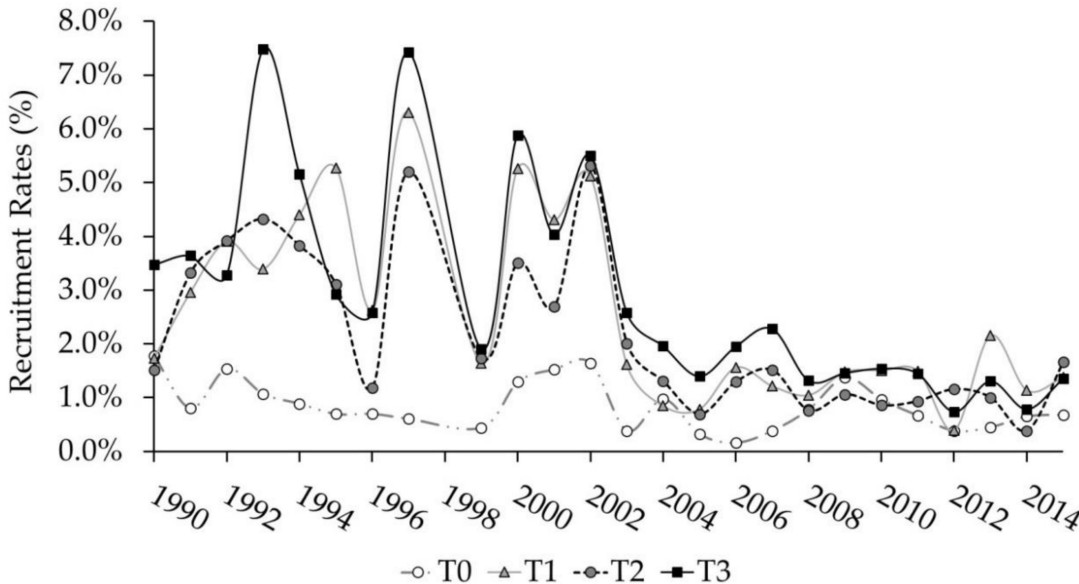

**Figure 2.** Dynamics in recruitment rate observed over 25 years of monitoring for all treatments studied.

The principal botanical families recruited during the study period, in the logged treatments, were as follows: Urticaceae for T1 at 13.6%, Fabaceae for T2 at 15.1% and Melastomataceae and Urticaceae for T3 at 17.6% each. In the treatment of higher intensity (T3), although the botanical families Melastomataceae and Urticaceae presented the same percentage in the number of trees recruited, the family Urticaceae was the one that had the greatest gain in stocking. The treatment with the greatest gain in stoking was the heavy intensity, followed by the low intensity and medium intensity. For the control treatment (T0), the botanical family that showed the greatest number of trees recruited was Lecythidaceae at 12.7%. However, it was the botanical family Fabaceae that presented the largest gains in stocking.

The species *Cecropia sciadophylla* Mart. (Urticaceae) had the highest total number of trees recruited in the T1 and T3 treatments at 7.2% and 9.8%, respectively. For treatment T2, the species *Croton matourensis* Aubl. (Euphorbiaceae) was the most recruited with 7.7% of the total number of individuals recruited. In the control (T0), the recruitment was greatest for the species *Protium hebetatum* D.C. Daly at 4.7%, and was the species that had the largest number of individuals recruited during the study period. In Table 4, it is possible to observe the five main species, of each treatment analyzed, that contributed most to the increase in basal area, volume, biomass and carbon.

**Table 4.** Species with the highest total number of: trees recruited (*N*); gain in basal area (AB), volume (VOL); biomass (BIO), carbon (CAR) over the duration of the study, and Ecological Group (EG): No Pioneer (NP), and Pioneer Species (PS).

| Treatment | Species | Botanic Family | N | AB (m$^2$) | VOL (m$^3$) | BIO (t) | CAR (t) | EG |
|---|---|---|---|---|---|---|---|---|
| | *Protium hebetatum* D.C. Daly | Burseraceae | 18 | 0.2 | 2.3 | 4.0 | 1.1 | NP |
| | *Eschweilera wachenheimii* (Benoist) Sandwith | Lecythidaceae | 12 | 0.1 | 1.5 | 2.5 | 0.7 | NP |
| T0 | *Eschweilera romeu-cardosoi* S.A. Mori | Lecythidaceae | 10 | 0.1 | 1.2 | 2.2 | 0.6 | NP |
| | *Rinorea paniculata* (Mart.) Kuntze | Violaceae | 9 | 0.1 | 1.1 | 2.0 | 0.6 | NP |
| | *Scleronema micranthum* (Ducke) Ducke | Malvaceae | 7 | 0.1 | 0.9 | 1.5 | 0.4 | NP |
| | *Cecropia sciadophylla* Mart. | Urticaceae | 83 | 1.2 | 18.5 | 30.8 | 8.7 | PS |
| | *Croton urucurana* Baill. | Euphobiaceae | 57 | 0.6 | 8.4 | 14.4 | 4.1 | PS |
| T1 | *Vismia* spp. | Hypericaceae | 54 | 0.5 | 7.7 | 13.2 | 3.8 | PS |
| | *Croton matourensis* Aubl. | Euphobiaceae | 50 | 0.5 | 7.3 | 12.6 | 3.6 | PS |
| | *Miconia* spp. | Melastomataceae | 33 | 0.3 | 4.3 | 7.5 | 2.1 | PS |
| | *Croton matourensis* Aubl. | Euphorbiaceae | 78 | 0.8 | 11.9 | 20.5 | 5.8 | PS |
| | *Cecropia sciadophylla* Mart. | Urticaceae | 64 | 0.8 | 11.8 | 20.1 | 5.7 | PS |
| T2 | *Byrsonima duckeana* W.R. Anderson | Malpighiaceae | 34 | 0.3 | 4.6 | 7.9 | 2.2 | PS |
| | *Miconia minutiflora* (Bonpl.) DC. | Melastomataceae | 27 | 0.2 | 3.7 | 6.4 | 1.8 | PS |
| | *Miconia* spp. | Melastomataceae | 25 | 0.2 | 3.3 | 5.8 | 1.6 | PS |
| | *Cecropia sciadophylla* Mart. | Urticaceae | 127 | 1.6 | 23.4 | 39.6 | 11.2 | PS |
| | *Miconia minutiflora* (Bonpl.) DC. | Melastomataceae | 64 | 0.6 | 8.8 | 15.2 | 4.3 | PS |
| T3 | *Bellucia grossularioides* (L.) Triana | Melastomataceae | 52 | 0.5 | 7.0 | 12.1 | 3.4 | PS |
| | *Miconia* spp. | Melastomataceae | 47 | 0.4 | 6.2 | 10.8 | 3.1 | PS |
| | *Croton matourensis* Aubl. | Euphorbiaceae | 45 | 0.4 | 6.7 | 11.5 | 3.3 | PS |

When comparing the number of trees recruited with the number of dead trees during the entire period of analysis, we observed that the recruitment was greater than mortality in all treatments where there was a reduction of basal area (Figure 3A). These values for recruitment were higher at 50.9%, 47.2% and 58.6%, respectively for T1, T2 and T3 than those for mortality. Within the control (T0), contrary to the logged treatments, we found that the mortality was greater than the recruitment (*p* = 0.248) by 14.4%.

The only logging treatment where there was balance between recruitment and mortality trees, was observed in T2 (*p* = 0.106). However, this was not the case observed for treatments T1 (*p* = 0.005) and T3 (*p* ≤ 0.0001). The loss in stocking caused by mortality was superior to that of replacement by recruitment: basal area (*p* < 0.0024) (Figure 3B), volume (*p* < 0.0023) (Figure 3C) and biomass (*p* < 0.0021) (Figure 3D) in all treatments.

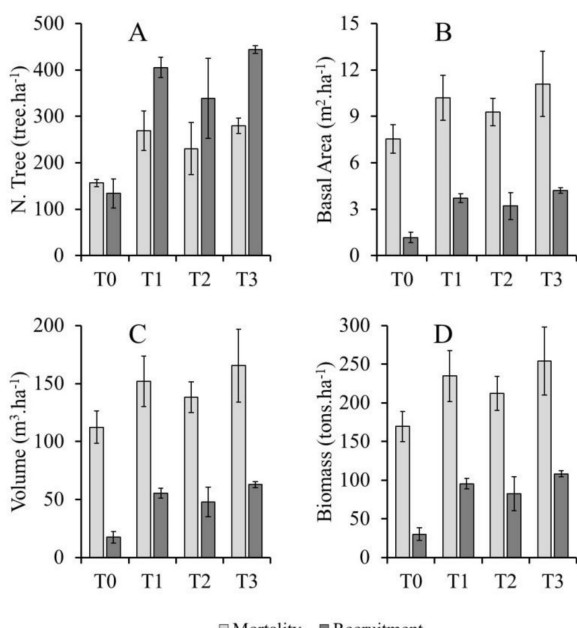

**Figure 3.** Dynamics of entry and exit of stocking during the monitoring period (*N* = 25 years): Number of trees (**A**); Basal Area (**B**); Volume (**C**); and Biomass (**D**).

### 3.3. Diameter Increments

During the monitoring period, the mean diameter increase was $0.25 \pm 0.02$ cm·year$^{-1}$ for T1, $0.26 \pm 0.02$ cm·year$^{-1}$ for T2 and $0.30 \pm 0.02$ cm·year$^{-1}$ for T3. For the control treatment (T0), the mean annual increment was $0.17 \pm 0.01$ cm·year$^{-1}$. The highest increases of mean annual increment occurred in 1992 in T1 with 0.34 cm·year$^{-1}$, in 1991 in T2 with 0.36 cm·year$^{-1}$ and in 2011 in T3 with 0.40 cm·year$^{-1}$ (the second highest peak of recruitment occurred in 1992 with 0.39 cm·year$^{-1}$) (Figure 4).

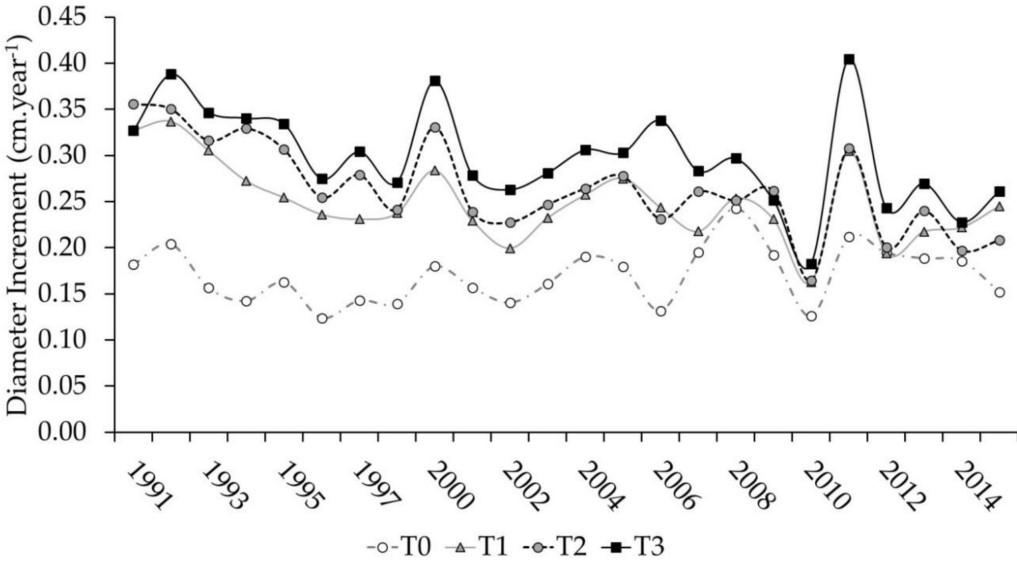

**Figure 4.** Diameter increment observed over 25 years of monitoring for all the treatments studied.

According to the ANOVA test, diameter increments varied significantly throughout the study period (ANOVA of Repeated Measures, $p < 0.0001$). The Control Treatment (T0) showed a significantly lower increase in diameter increment compared to the other treatments (Tukey post hoc, $p \leq 0.0001$). As for T1 and T3, we observed that it is very unlikely that the values obtained were statistically similar ($p = 0.010$). This same result was also observed between the values of the T2 and T3 Treatments ($p = 0.041$). However, it is very likely that the increases obtained between treatments T1 and T2 are statistically similar ($p = 0.753$).

The greatest increase to diameter increment occurred shortly after logging, decreasing over time for T1 ($r = -0.48$, $p = 0.000$), T2 ($r = -0.67$, $p = 0.000$) and T3 ($r = -0.45$; $p = 0.000$), that is, in treatments that experienced a reduction in basal area (Figure 5). However, in the Control Treatment (T0), the diameter increments of the remaining trees increased with the passage of time ($r = 0.22$, $p = 0.060$). Based on forest growth monitoring, there was a positive ($r = 0.93$) and highly significant correlation ($p < 0.0001$) with the intensity of the logging.

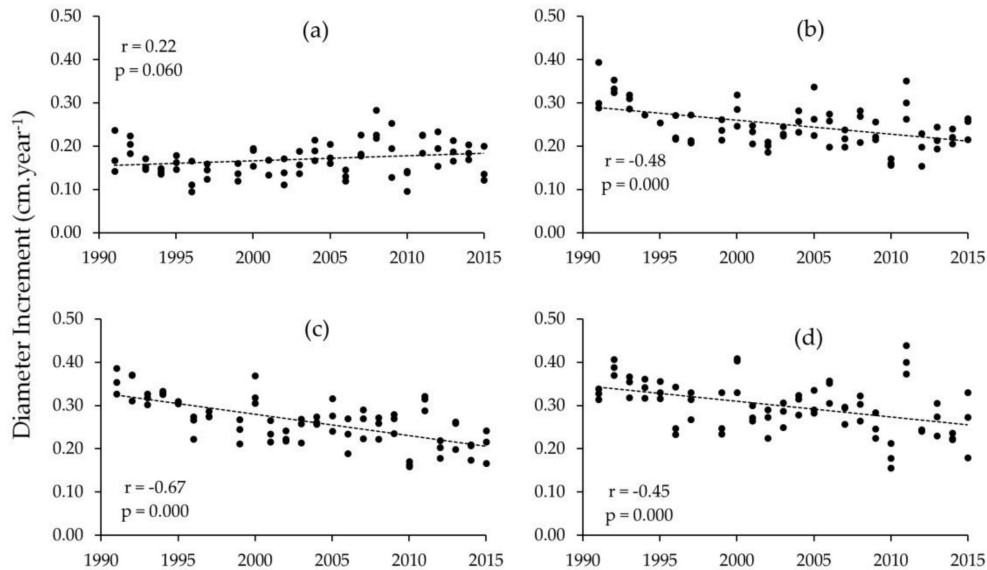

**Figure 5.** The graph shows the diameter increase for each treatment analyzed: Control (**a**); Low Intensity (**b**); Medium Intensity (**c**), and Heavy Intensity (**d**). It is possible to observe that in the plots where there was silvicultural intervention the diameter increment decreases with the passage of time. However, in the control plot the increment increased during the study period.

The species that obtained the highest averages of increment growth were: *Tachigali glauca* Tul. at $0.80 \pm 0.41$ cm·year$^{-1}$ in T1; *Pseudopiptadenia suaveolens* (Miq.) J.W. Grimes at $1.10 \pm 0.02$ cm·year$^{-1}$ in T2; and *Vochysia densiflora* Spruce at $0.89 \pm 0.31$ cm·year$^{-1}$ in T3. For the Control Treatment (T0), the species that obtained the highest average increment during the studied period was *Ocotea floribunda* (Sw.) Mez with $1.11 \pm 0.22$ cm·year$^{-1}$. The species whose results showed the lowest average increment were: *Cordia* sp. at $0.04 \pm 0.03$ cm·year$^{-1}$ in T1; *Amphirrhox longifolia* (A.St.-Hil.) Spreng. at $0.06 \pm 0.01$ cm·year$^{-1}$ in T2 and $0.04 \pm 0.03$ cm·year$^{-1}$ in T3. The species with the lowest average increment obtained in T0 was *Pachira macrocalyx* (Ducke) Fern.Alonso with $0.03 \pm 0.00$ cm·year$^{-1}$.

The botanical families with the highest mean increments were: Malpighiaceae at $0.51 \pm 0.09$ cm·year$^{-1}$ in T1; Urticaceae at $0.59 \pm 0.07$ cm·year$^{-1}$ in T2; and Malpighiaceae at $0.51 \pm 0.09$ cm·year$^{-1}$ in T3. For T0 the botanical family that obtained the highest rates of annual average increment was the Urticaceae with $0.46 \pm 0.12$ cm·year$^{-1}$. The botanical families with the lowest mean increment were: Siparunaceae at $0.07 \pm 0.02$ cm·year$^{-1}$ in T1; $0.09 \pm 0.04$ cm·year$^{-1}$ in T2; and Sapindaceae at $0.10 \pm 0.04$ cm·year$^{-1}$ in T3. For the Control Treatment (T0) the botanical family with the smallest increase was Icacinaceae at $0.05 \pm 0.04$ cm·year$^{-1}$.

## 4. Discussion

The results demonstrated that the application of different logging intensities to the forest altered the dynamics, increasing mortality rates, recruitment and adding to the increment of the remaining trees. The increase in the mortality rate, shortly after logging, may have occurred due to the damage caused by the logging operations. Similar results were observed as the current study where the high mortality rates mainly occurred after forest intervention (3.71% year$^{-1}$) stabilizing five to nine years after logging (1.20%) [35].

According to one study carried out in the Amazon, selective logging caused on average approximately 24.5% damage to the residual forest stand [36]. The main damage caused to the remaining trees were: loss of crown (12%), damage from tractors (11%) and bark damage (3.1%) [16]. Due to these factors, there was a decline in forest stocking. For each commercial tree logged, approximately 0.5 m$^2$ of basal area was severely affected by the logging activity for individuals with DBH $\geq$ 10 cm [15]. These studies corroborate the results found in our research and explain the

increase in the mortality rate shortly after logging. The findings suggest that this increase is related to the damage caused to the remaining forest.

The mortality rates for the different treatments obtained in this study (Table 2) were lower than the rates found by [35]. These authors analyzed the effect of different levels of timber extraction, combined with silvicultural treatments on forest dynamics, twenty years after selective logging. For the above mentioned study, the authors found the following mortality rates: 2.15% year$^{-1}$ for light treatments, 2.74% year$^{-1}$ for the moderate treatments and 2.60% year$^{-1}$ for heavy treatments.

The mortality rate found in the control treatment (T0) was approximately 50% lower than the value found by [37] in unlogged forests. In the study of [37], carried out in forests under forest management in the eastern Amazon, state of Amapá, the mortality rate encountered was 1.82%, while our results presented a rate of 0.99% $\pm$ 15%. In the mainland tropical rainforests of the Amazon, mortality rates are between 1% and 2% year$^{-1}$ [3,16,38–42]. Recruitment rates in unlogged old-growth forests of the Amazon are close to the mortality rates, which are between 0.9% and 1.8% year$^{-1}$ [16,38,43,44], displaying the dynamic equilibrium of undisturbed forests.

Over the years, we have observed that the mortality rate has been declining and stabilizing for about 10 years, in this study. However, after the first decade, the mortality rate did once again increase, especially between the years of 2005 and 2011 in all treatments. During this period, mortality rates reached higher levels, in some cases, to the values found shortly after logging: 3.8% in T1, 2.5% in T2 and 4.1% in T3. In the case of T3, this value was the second highest peak in the mortality rate observed throughout the monitoring years. It is important to note that during the same period (2005 and 2010), two of the most severe droughts recorded in the last hundred years occurred throughout the Amazon basin [45,46]. These droughts made it possible to directly assess the sensitivity of the tropical forest to water deficit [46,47].

In 2005, near the area of our study, a natural phenomenon known as a convective storm, also known as a "downburst," occurred. This event caused the mortality of 542 $\pm$ 121 million trees in the entirety of the Amazon basin [48]. According to [12], extreme weather events that cause a massive amount of blowdowns contribute to the increase in tree mortality. Climate change may increase rainfall intensity in the Amazon, which in turn can produce more blowdowns in the forest associated with convective storms [48]. In this way, the occurrence of these events, in shorter and shorter periods, may have important impacts on the forest dynamics. These results suggest that the increase in tree mortality rates in our study during this 10-year period may have been influenced by the occurrence of these events.

A second hypothesis for the increase in the mortality rate can be explained by the emergence of pioneer species soon after the logging operations (Table 4). These species are characterized by growth and occupancy in disturbed environments in a rapid manner and short life cycle [49,50]. In this sense, for these treatments (T1, T2 and T3), one of the hypotheses in the increase in mortality frequency in the last study periods is due to the effects of senescence of these species. However, for the Control Treatment (T0), in which there was no logging, we observed that the mortality rate also increased in the last year of the inventory (1.6%). This rate was even higher than the values found in the treatments in which there was a reduction of basal area for the same year (1.2% in T1, 0.7% in T2 and 1.4% in T3).

Studies indicate that the unlogged forests of the Amazon are becoming more dynamic over time, raising tree mortality and turnover rates [24,25,51]. This increase may be associated with constant climatic variations, mainly from the change in rainfall and dry season intensity [47,52–54]. The combination of less rainfall with increasing temperature increases the probability of tree mortality, mainly due to fires or water stress [20].

A third hypothesis to explain the increase in mortality in recent years is that the prolonged actions of climate and environmental changes stimulate forest growth and productivity in the Amazon [24]. This has caused factors such as competition for light, nutrients and physical space between species to increase as well, thus reducing these resources which can lead to tree death [55]. It is noteworthy that the increase in tree mortality may be the result of the interaction of all these factors, since they

do not occur in isolation. However, this statement can only be tested with longer, standardized and continuous monitoring periods.

As for the increase in the recruitment rate of trees, soon after logging, it may have been caused by the removal of larger trees (logging) and the high mortality of the damaged trees that remained. These effects open the forest canopy, create clearings, increase sun light exposure and diminish competition, which in turn, promote the recruitment of new trees. In these areas, pioneer species, characterized by rapid growth and occupation of disturbed environments, dominate the clearings in a very short period of time [49,50].

Recruitment intensity is directly linked to the size of the clearing [56]. A prior study that was carried out in the same area as this study, that evaluated the recruitment of pioneer species, obtained the following annual recruitment rates: $1.70 \pm 0.39\%$, $1.40 \pm 0.45\%$ and $2.32 \pm 0.53\%$, respectively, for T1, T2 and T3 treatments [57]. These same authors observed that the maximum number of trees recruited, considering all the species, happened 10 years after logging. This statement corroborates our results, in which it is possible to observe that the recruitment rate decreases considerably after 15 years after logging (Figure 2). This is due to the closure of the forest canopy and the maximum occupancy of the site.

Moreover, our results showed that recruitment increased in the period following the peaks in mortality, shortly after logging (Figures 1 and 2). This same result was noted by [25], where recruitment of trees was strongly predicted by mortality from the previous year. Additionally, the recruitment increased in the period following peak mortality, creating a time lag between mortality and recruitment. Hence, it is possible to predict that in the coming years the recruitment rate will again increase due to the high mortality rate found in the last years of observation.

It was also possible to observe that the lower recruitment and mortality rates were found in treatment T2, which was followed by the T1 and T3 treatments. These results differ from the logging intensity, in which, theoretically, a direct correlation between the operating intensities and the recruitment rate should follow. However, the form of distribution, as well as the size of the gaps of the different treatments, was smaller in T2 [58]. Consequently, this means that logging caused a smaller impact on the residual forest canopy in Treatment T2. This observation may be the explanation for the reduced recruitment in T2, that is, the smaller canopy openings lead to diminished recruitment of trees.

Similar studies found lower recruitment estimates: light logging intensity estimated 1.68% year$^{-1}$ recruitment, medium intensity = 2.15% year$^{-1}$ and high intensity = 2.41% year$^{-1}$ [35]. The recruitment rate for treatment T3 (Table 2) is similar to the values of the studies carried out on the Tapajós National Forest (3.1% year$^{-1}$) 11 years after a high logging intensity [59].

When considering exclusively the number of trees entering and leaving the research site, we found that the mortality and recruitment rates in the T0 treatment indicate that the Central Amazonian forests are in a dynamic equilibrium ($p = 0.19$). This means that trees that leave the system (mortality) are continually replaced by new individuals (recruitment). This balance was also observed in other similar studies [39,49,60,61].

However, when analyzing the losses and gains in stocking (basal area, volume and biomass), it is possible to observe that losses (Figure 3), caused by mortality, were higher in all treatments analyzed. This is because tree mortality affects all individuals, independent of the DBH class. Recruitment usually occurs only in the initial DBH classes of $\geq 10$ cm. When a tree larger than 50 cm DBH dies, for example, the loss in biomass is approximately 35 times greater than when a tree is recruited. Although the number of trees recruited is approximately 40% greater than mortality, the loss of basal area, volume and biomass stocking is much larger than the recruitment gain.

Our findings on DBH increment over time showed similar results to [41], which were in the Tapajós and Jari regions, where the authors verified that the diameter increment in mature logged forests varies from 0.2 to 0.3 cm·year$^{-1}$ and in unlogged forests was 0.1 cm·year$^{-1}$. For the Jari region, the average growth rate was 0.4 cm·year$^{-1}$. In the Tapajós National Forest the remaining trees (DBH $\geq 5$ cm),

when considering all species, had an average diameter increment no higher than 0.30 cm year$^{-1}$ within a 16 year interval (1981 to 1997) after selective logging [62]

The average diameter increment, shortly after logging, could be explained by the removal of the trees through selective logging, natural mortality and/or mortality due to logging. Forests gaps promoted the increase of light exposure, decreased competition for nutrients and water, which enabled the remaining trees to invest in growth. Soon after the disturbance, pioneer trees colonized and established themselves within the site quickly. This process of succession had a direct influence on the value of the final average increment. The gap opening in the forest canopy and the rapid growth in height and diameter of pioneer species resulted in the greater increases in the logged treatments. This information is evident when we compare the results of the average increments of the forests that were harvested (ranging from 0.25 to 0.30 cm·year$^{-1}$) with a forest that was not harvested (0.17 cm·year$^{-1}$).

The treatment, consisting of a 75% reduction of the basal area (T3), was the one that contributed to the highest average values of the overall diameter increment. This result differed from the results found by [35]. This author, evaluating the dynamics of the remaining vegetation that was subjected to three different logging intensities, combined with four levels of basal area reduction, observed that the treatment which suffered the highest intensity of basal area reduction was not the treatment with the highest annual increment rate.

The lowest mean diameter increment was registered in the year 2010 (Figure 4): 0.16 cm·year$^{-1}$ in T1, 0.16 cm·year$^{-1}$ in T2 and 0.18 cm·year$^{-1}$ in T3. In the same year, T0 treatment recorded the second lowest mean of increment growth at 0.13 cm·year$^{-1}$. It is important to note that in this same year, the state of Amazonas experienced the largest drought recorded since 1953. In July 2009, an El Niño, out of season, was formed in the Pacific in which the peak occurred in October of the same year and extended until July 2010.

According to [12], the increase in diameter shows a positive correlation ($r > 0.70$) and highly significant correlation ($p < 0.001$) with rainfall. Hence, it is possible to affirm that the diameter increases of the remaining trees was affected by the drought of 2010. However, these results differ from [25], in which they evaluated the dynamics of a forest monitored from 1981 to 2003. The authors observed that tree growth was greatest during periods of drought.

## 5. Conclusions

During the monitoring period, many species entered the post-logging site, and diverse structural changes occurred in the dynamic of the remaining trees. However, most of these tree species are not considered commercial species. In addition, many commercial species succumbed to mortality, or they were removed or damaged from the logging operations. Our study suggests that logging affects the dynamics of the residual forest, which increases recruitment rates, mortality and diameter increment over the following decades, while contributing to changes in species composition. In this sense, it is very difficult to infer, in short-term studies, without standardization of measurements or long-term research intervals, on future cutting cycles. Despite 25 years of annual monitoring and 30 years after logging, the study period of this work is considered short when trying to infer growth patterns about the sustainability of forest management in the highly complex dynamics of tropical forests. In this context, it is important to continue the long-term monitoring and the continuity of the studies of the environmental responses to logging and forest management.

**Author Contributions:** M.R.M.A. collected the experimental data, analyzed and wrote the manuscript. All co-authors participated equally in conceptualization, investigation, data analysis and manuscript preparation and review.

**Funding:** This research was funded by CAPES (Coordenação de Aperfeiçoamento de Pessoal de Nível Superior) (Process: 973613).

**Acknowledgments:** The authors would like to acknowledge INCT-Madeiras da Amazônia for the technical support, the ZF-2 Staff and the Forest Management Laboratory of INPA. Many thanks to Tatiana Gaui for the instruction on Dendrology, and Bruno Gimenez and Daniel DeArmond for the paper revision.

**Conflicts of Interest:** The authors declare no conflict of interest.

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
