# Peer review of "Dynamics of Tropical Forest Twenty-Five Years after Experimental Logging in Central Amazon Mature Forest"

_forests, doi:10.3390/f10020089_

Round 1

Reviewer 1 Report

The introduction can start at line 84. Everything that came for that is not relevant for the paper. However, it is necessary to explain in the introduction why the various attributes of the tree vegetation that were measured after treatments are relevant to measure. Why is it relevant to measure mortality, recruitment, diameter increase after interventions. What do these attributes tell us; why is it relevant to observe them. That is not explained explicitly in the introduction nor methods, but it should be. 

The results section has for each of the variables (mortality, recruitment, diameter) one or several paragraphs in which results are spelled out. These paragraphs are tedious to read. All that information is or can go into tables and the results sections should point out only the most relevant points. 

An academic paper should use full binomials and authors for plant species. I.e. Cecropia sciadophylla  should be  Cecropia sciadophylla  Mart.

Author Response

Response to Reviewer 1 Comments

Point 1: The introduction can start at line 84. Everything that came for that is not relevant for the paper

Response 1: The authors believe that, when dealing with forest management that is carried out mainly in Amazonian forests, it is important to describe the role of the forest in relation to biodiversity, climate, hydrological cliclo and carbon. Also, it is important to note that forest management helps to keep tropical forests standing, unlike other forms of land use that are practiced in the Amazon and other tropical regions.

Point 2: However, it is necessary to explain in the introduction why the various attributes of the tree vegetation that were measured after treatments are relevant to measure. Why is it relevant to measure mortality, recruitment, diameter increase after interventions. What do these attributes tell us; why is it relevant to observe them. That is not explained explicitly in the introduction nor methods, but it should be.

Response 2: We believe that this information was included in the lines: 84-94

Point 3: The results section has for each of the variables (mortality, recruitment, diameter) one or several paragraphs in which results are spelled out. These paragraphs are tedious to read. All that information is or can go into tables and the results sections should point out only the most relevant points

Response 3: Done. These paragraphs have been restated

Point 4: An academic paper should use full binomials and authors for plant species. I.e. Cecropia sciadophylla  should be  Cecropia sciadophylla  Mart.

Response 4: Done. For all the species mentioned, the necessary changes were made.

Reviewer 2 Report

Overall, this was an interesting study.  Please see the attached comments and suggestions.

Author Response

Point 1: Abstract - Lines 18-23 Rather than report all of the values, report how the treatments were different (e.g., which had the highest/lowest mortality/recruitment and the percentage difference). In addition, how was the recruitment different among the treatments? I think this is part of the dynamism that I was looking for.

Response 1: after the suggestions: The highest mean annual mortality rates occurred in the T3 treatment (1.82% ± 0.38). The same occurred in the mean annual recruitment rates (2.93% ± 0.77) and mean annual diameter increments (0.30 ± 0.02 cm). The loss in stocking caused by mortality, was superior to that of replacement by recruitment. The results showed that selective logging altered the natural dynamics of the forest, increased morality rates, recruitment and increased growth rates of the residual trees.

 Point 2: Introduction - L97 Change exploration to “exploitation” – I think this is what was meant to be said.

 Response 2: Done

 Point 3: Materials and methods - 123-128 Based on the text, it looks as if there were two replicates of each treatment, is this correct? If so state that effectively there were two replicates of each

 Response 3: I’ve changed the text for a better understanding. There are three replicates of each treatment,

 Point 4: Materials and methods -  Table 1. caption – change “tress” removed to “trees” removed

 Response 4: Done.

 Point 5: Materials and methods - L 188-193 Did the data meet the assumptions to perform the tests? What the significance level (I assumed it was 0.05)? What statistical software was used?

 Response 5: after the suggestions: When the ANOVA test presented significant statistical results (p < 0.05), a post hoc test of Tukey was applied. Linear regressions were also adjusted for diametric increment across the years for all remaining trees of each treatment. In addition, a t-test was performed between the recruitment and mortality stocks of each treatment and presented in Figure 3. All analysis were performed on Systat 13.2 free versions.

 Point 6: Materials and methods - L 193 Change “Figure 03” to “Figure 3”

 Response 6: Done

 Point 7: Results - L 199-201 These are values that are in the table, why not point out how different the rates of mortality were among the different treatments? I think it would highlight how the different treatments affect what was seen

 Response 7: After suggestions: These values represented an annual average mortality rate per hectare no proportional of different cutting intensities, for example, the medium intensity treatment showed lower mortality rates values.

 Point 8: Results - Figure 1 Please enlarge this figure, it is difficult to make out at 100% view, also remove the title “Mortality Rates”.

 Response 8: Done

 Point 9: Results - Lines 229-248 These paragraphs are merely repeating the tabular information, why not indicate how the botanicals were different among treatments as well as the overall trends in the data itself – it is extremely difficult getting through all of the values in the text.

 Response 9: Part of the text has been modified, as suggested by the reviewer

 Point 10: Results - Table 3 Change “Especie” to “Species”

 Response 10: Done

 Point 11: Results - Lines 254-277 As with the edit suggested above – it seems that there are a substantial number of values to get through – I think the results would be better served pointing out where the differences were and how large they were. In addition, the interaction between time recruitment and treatment should be detailed first, looking at main effects should be reported second (it would be difficult attributing what was seen to the main effects due the significant interaction).

 Response 11: Part of the text has been modified, as suggested by the reviewer

 Point 12: Results - Figure 2 Please enlarge this figure, it is difficult to make out at 100% view, also remove the title “Recruitment Rates”.

 Response 12: Done

 Point 13: Results - Lines 280-300 Please see the comments for lines 229-248. I would rather know how different the treatments were, rather than see the values here.

 Response 13: some not-so-relevant information has been deleted. It was inserted in the document text: The treatment with greatest gain in the stocks was heavy intensity, followed by the low intensity, and medium intensity.

Point 14: Results - Table 4 Change “Especie” to “Species”

Response 14: Done

Point 15: Results - Lines 304-314 Both paragraphs seem to report the same results

Response 15: the first paragraph is the result on the number of trees recruited and second paragraph is the result on stocks in basal area, volume and biomass.

Point 16: Results - L 307-309 Was the difference in the T0 significant between mortality and recruitment?

Response 16: The statistical significance between mortality and recruitment is now presented (p = 0.248)

Point 17: Results - Lines 310-311 So these findings were not significant?

Response 17: Yes (T0: p = 0.248)

 Point 18: Results - Figure 03 Change to Figure 3. What do the light and dark columns represent? Do the columns represent the entire study period? N = ? In addition, there is nothing in the text that mentions panels B-D. How do these panels relate to the mortality/recruitment?

Response 18: Done. The information is inserted in the text.

Point 19: Results - Lines 319-323 This paragraph is a bit confusing. I find that the more interesting trends are in the fact that T0 had the lowest increment growth, on average, whereas T3 had the highest. Because we need to know more about the effects of forestry practice on growth, I think this would be more informative.

Response 19: these are just the general results of the diametric increase of each treatment.

Point 20: Results - L 323 Change Figure 04 to Figure 4

Response 20: Done

Point 21: Results - Lines 335-336 But the incremental change was not significant at α = 0.05, and should be stated as such.

Response 21: this value is shown in line 327-328

Point 22: Results - L 334 Be sure the p-values reported in the text are the same as those reported in the figure.

Response 22: Done

Point 23: Results - L 335 Change Figure 05 to Figure 5.

Response 23: Done

Point 24: Discussion - L 361 Please indicate the author’s name rather than the reference number.

Response 24: Done

Point 25: Discussion -  L 362 How much time had elapsed in [46] study? It would be interesting and maybe important to know how long it took for the stabilization in the other study to occur.

 Response 25: Done

Point 26: Discussion - Lines 369-370 Rather than submit “this means”, state that “the findings suggest”. This gives the authors a little wiggle room.

Response 26: Done

Point 27: Discussion - L 427 “Directly linked”

Response 27: Done

Point 28: Discussion - L 427 Please indicate the author’s name rather than the reference number.

Response 28: Done

Point 29: Discussion - L 432 What was the “certain period”? State the timeframe here

 Response 29: Done

Point 30: Discussion - L 448 Please indicate the author’s name rather than the reference number

Response 30: Done

Point 31: Discussion - L 467 Please indicate the author’s name rather than the reference number.

 Response 31: Done

Point 32: Discussion - L 476 Which species are “pioneers” in this system? In tables 3 & 4 several species are mentioned as well as the text L339-248 & L280-300. Perhaps this is the area where the species lost should be mentioned as pioneers vs. late successional species. I think it would help your arguments as to the gap openings and the recruitment of the pioneers.

Response 32:  I believe this is presented in the discussion of the work (lines 445-449 and 477 481

Point 33: Discussion  - Lines 482-485 Why do you think your findings were so different?

Response 33:  I believe this is presented in the discussion of the work (lines 445-449 and 477 481

Point 34: Discussion - Lines 486-496 These two paragraphs should be combined since they are both discussing climate effects

Response 34:  I belive this is contemplated  in the text. It is important to note that in this same year, the state of Amazonas experienced the largest drought recorded since 1953. In July 2009 an El Niño, out of season, was formed in the Pacific in which the peak occurred in October of the same year and extended until July 2010. The increase in diameter shows a positive correlation (r > 0.70) and highly significant correlation (p < 0.001) with rainfall. Hence, it is possible to affirm that the diameter increases of the remaining trees was affected by the drought of 2010

Round 2

Reviewer 1 Report

I have added sticky notes to the latest manuscript. I hope the authors find these relevant and are able to address the comments in them.

I do insist on revising the introduction. The paper is about the impact of different logging intensities on neotropical forests. The introduction should be limited to that topic and not dwell on other issues that are not central to the paper. In the Discussion the authors could add some ideas on what the results imply for neotropical forest management, and possibly make a link to other societal demands on forests.

Author Response

Response to Reviewer 1 Comments – Round 2

Point 1: What is the difference? Why would you estimate qualitatively commercial volume stocking if you have estimated it qualitatively?

Response 1: Maybe your question is: quantitatively and qualitatively, right?

So, the main goal of forest inventory is to estimate quantitatively and qualitatively stocks, mainly, to commercial species, i.e. which commercial species occur and how much (by volume, biomass, for example) exists in the area.

Point 2: The subheadings 2.4.1 - 2.4.5 can be deleted. Equally, the subheading 2.5. can be deleted.

Response 2: Done

Point 3: This paragraph needs improvement. Statements like "... it is virtually certain that ..." is inappropriate language in an academic article.

Response 3: Done. These paragraphs have been restated

Point 4: This is something counter intuitive and would require suggestion of a hypothesis on why that is the case, or else at least a statement that it cannot be explained.

Response 4: These were modifications suggested by another reviewer

Point 5: Is this one unidentified species, or are these several species. If one species, the word genus should be replaced by species. If more species, the sp should be replaced by spp.

Response 5: Done.

Point 6: In the table there are several species under each genus name (spp), so there are more than five species. If that is not the case, than spp, should be changed into sp.

Reponse 6: There are several species of the corresponding genus

Point 7: Better to be consistent and always refer to treatments as T0, T1, T2 and T3.

Reponse 7: Done

Point 8: Same as above. In several cases more than species are suggests (spp). If these represent one single species, that should be change to sp. If these represent more spcies, than the mentioning of five species in this phrase should be changed.

Response 8: There are several species of the corresponding genus

Point 9: to change “logged” to loggin ; “diameter decreases” to diameter increment decreases

Response 9: Done
